# Valorisation, Green Extraction Development, and Metabolomic Analysis of Wild Artichoke By-Product Using Pressurised Liquid Extraction UPLC–HRMS and Multivariate Data Analysis

**DOI:** 10.3390/molecules27217157

**Published:** 2022-10-22

**Authors:** Stefania Pagliari, Ciro Cannavacciuolo, Rita Celano, Sonia Carabetta, Mariateresa Russo, Massimo Labra, Luca Campone

**Affiliations:** 1Department of Biotechnology and Biosciences, University of Milano-Bicocca, Piazza Della Scienza 2, 20126 Milano, Italy; 2Department of Pharmacy, University of Salerno, Via Giovanni Paola II 132, Fisciano, 84084 Salerno, Italy; 3Safety and Sensoromic Laboratory (FoCuSS Lab), Department of Agriculture Science, Food Chemistry, University of Reggio Calabria, Via dell’Università 25, 89124 Reggio Calabria, Italy

**Keywords:** artichoke leaves by-product, pressurised liquid extraction, phytochemical analysis, antioxidant activities, multivariate analysis

## Abstract

Valorisation of food by-products has recently attracted considerable attention due to the opportunities to improve the economic and environmental sustainability of the food production chain. Large quantities of non-edible parts of the artichoke plant (*Cynara cardunculus* L.) comprising leaves, stems, roots, bracts, and seeds are discarded annually during industrial processing. These by-products contain many phytochemicals such as dietary fibres, phenolic acids, and flavonoids, whereby the most challenging issue concerns about the recovery of high-added value components from these by-products. The aim of this work is to develop a novel valorisation strategy for the sustainable utilisation of artichoke leaves’ waste, combining green pressurised-liquid extraction (PLE), spectrophotometric assays and UPLC–HRMS phytochemical characterization, to obtain bioactive-rich extract with high antioxidant capacity. Multivariate analysis of the major selected metabolites was used to compare different solvent extraction used in PLE.

## 1. Introduction

The traditional model economy based on “take–make–dispose”, called linear economy, of our consumption and resources use pattern has led to several environmental problems making it unsustainable. Nowadays, growing attention has been paid worldwide to moving beyond the linear economy, to a new concept of “take–make–use–recycle”, the circular economy, in which the waste of a process is not directly discarded but becomes resources for new products and application. In this context, particular attention has been paid to the agro-industry, in which its food wastes can take several stages of the supply chain [1]. A large amount of waste produced by the food industry, in addition to being a great loss of valuable materials, also raises serious management problems, both from the economic and environmental points of view. Considering that agro-industrial by-products have the potential to be reused into other production systems, many of these residues should be increasingly regarded as potential resources [1]. In fact, from the treatment of agri-food by-products, it is possible to recover compounds with a high biological and economical value which could generate economic and environmental benefits.

Artichoke (*Cynara cardunculus* L.) is a herbal plant belonging to the Asteraceae family. It is typical and native to the Mediterranean area, grows mainly in temperate areas with hot summers and cool winters, prefers dry and sandy soils, and is also able to tolerate a moderate salinity [2,3,4]. The *C. cardunculus* species has numerous varieties, the main ones being *C. cardunculus* var. *scolymus*, corresponding to the domesticated artichoke using for food crops, and the *C. cardunculus* var *sylvestris*, that grows spontaneously in southern Italy where it is traditionally used [4]. *Cynara cardunculus* L. var. *sylvestris*, also known as wild cardoon, is a non-domesticated robust perennial plant characterized by a rosette of large spiny leaves and branched flowering stems. It is native to the Mediterranean basin (continental zone and isles), and the species is also naturalized in North and South America and in Australia, where it grows in a variety of habitats, colonising dry and undisturbed areas [5]. The varieties of artichoke differ both morphologically and in the content of secondary metabolites [6,7,8]. The main biomolecules contained in the artichoke are caffeoylquinic acids, such as cynarin and chlorogenic acid, flavone glycosides, inulin sugars, and minerals [9,10,11]. They contribute to the health-promoting effects on humans and there are known, different studies that show the use of artichoke as antilipidemic, diuretic, anti-inflammatory, and hepatoprotective [12], making it a “functional food” with interesting pharmacological and nutraceutical activities [13]. Italy is one of the main producing countries of artichoke, with an annual production, in 2020, of about 367.080 tonnes (FAO 2020) (https://www.fao.org/faostat/en/#data/QCL/visualize (accessed on 24 September 2022)). Large quantities of non-edible parts of the artichoke plant comprising leaves, stems, roots, bracts, and seeds are discarded during industrial processing which represent about 70–80% of the total plant biomass [11,14,15]. These biowastes contain a high moisture content making them susceptible to microbial growth and consequently may cause environmental contamination. However, these by-products are high in many phytochemicals similar to those of its edible flower heads including phenolic compounds, [14,15,16], inulin [17], and dietary fibre [18]. Based on interest in renewable resources of pharmaceutically active biomolecules, many investigations have been conducted on recycling and the optimisation of the downstream processes for maximal exploitation of artichoke biowastes [10,19,20,21,22,23]. However, most of these studies have focused their attention mainly on the bracts, giving little importance to the leaves which, although to a lesser extent, represent a huge waste. Valorisation of *C. cardunculus* L. biowastes represents opportunities for applications in the food industries such as functional and health-promoting ingredients.

In order to enhance the recovery of bioactive molecules from artichoke by-products, it is necessary to use techniques with low environmental impact and reduced production costs. Therefore, the search for increasingly efficient and environmentally sustainable extraction methods, that can mitigate the limitations associated with conventional extraction methods, has a great deal of interest in the food industry [24]. Indeed, green extraction techniques, along with the use of eco-friendly solvents, are required for the preparation of safe phytoextracts with nutraceutical interest, as is the chemical investigation of compounds with health-promoting effects, allowing the efficiency of the extraction process to be improved [25]. In this research, two different alternatives to conventional extraction methods were used: pressurized liquid extraction (PLE) and ultrasound assisted extraction (USAE) techniques. PLE uses high temperatures and pressures to reduce the viscosity of solvents improving the permeability and solvent-extractive capacity to increase the performance of the extraction as yield [26], specificity and reproducibility but also to reduce extraction time and organic solvent consumption. Moreover, solvents such as hydroalcoholic mixture or water are non-toxic and eco-friendly, hence these solvents are largely used for the selective extraction of polar compounds from food matrices.

Previous studies indicated a possible utilisation of this species to produce lignocellulos for energy, for inulin extraction [27]. One paper evaluated the content of phenolic acids, flavonoids, and their antioxidant activity of wild *Cynara cardunculus* L. using maceration as an extraction method [28]. To the best of our knowledge, there are no experimental works previously published concerning the utilisation of pressurised liquid extraction as a green technique for the recovery of bioactive compounds from wild cardoon by-products.

Therefore, the purpose of this work was to investigate the possibility of using a low environmental-impact green extraction technique to valorise a food product and at the same time obtain a comparison of the different extraction condition used using a multivariate approach.

The aim of this study was develop a “green” extract from *Cynara cardunculus* var. *sylvestris* by-product and evaluate their chemical profiles and radical scavenging activity. The comparison of profiles obtained by LC-ESI Q-TOF (HRMS) analysis was followed by quantitative analysis (UPLC–UV) of the main compounds (caffeoylquinic acids and flavones) for their antioxidant activity. Total phenol content (TPC) and antioxidant activity of the extracts were measured by Folin–Ciocalteu and the 1,1-diphenyl-2-picrylhydrazyl (DPPH) spectrophotometric assays, respectively. Finally, an approach based on targeted metabolomics analysis using multivariate data analysis (MVDA) was used to highlight differences among extraction conditions and evaluate the possible organic solvent reduction with the PLE technique.

## 2. Results and Discussion

### 2.1. Characterisation of Ultrasound-Assisted Extraction by UPLC–HRMS Analysis

The artichoke leaves’ by-products were extracted by USAE technique, using different solvents (H_2_O, EtOH, and EtOH50%) to select the conditions to obtain a representative composition of the sample metabolites to be used for qualitative characterisation by UPLC–HRMS. The extraction yields obtained using H_2_O EtOH and EtOH50% as the solvents were 18, 6 and 23%, respectively. Furthermore, the results of the spectrophotometric tests (Folin–Ciocalteu and DPPH) reported in Table 1 and the comparison of the USAE chromatographic profile (Figure 1) suggested that EtOH50% was the condition which provided the most representative metabolite profiles.

For these reasons, the extract obtained with EtOH50% was selected for characterisation by UPLC–HRMS; this data is in accordance with literature that often reported the hydroalcoholic mixture as the better condition for the recovery of polyphenols from artichoke samples [29,30]. The UPLC conditions were optimized to obtain maximal chromatographic resolution and higher MS signal response. In the mass spectrometry analysis, both negative and positive ionization modes were tested, and results showed that most of the compounds exhibited higher responses in negative ion mode, thus were selected for characterization purpose. Identities of compounds were assigned by comparing their retention times (Tr), UV/vis signals at selective wavelengths, accurate mass, molecular formula, and diagnostic fragmentation patterns MS/MS spectra with reference standards whenever available or combined with chemo-taxonomic data reported in the literature and databases. Specifically, Table 2 shows the 25 characterized analytes, belonging to the classes of hydroxycinnamic acids and flavonoids.

### 2.2. Optimisation of PLE Conditions

To evaluate the possibility of the use of *C. cardunculus* subsp. *scolymus* by-products as inexpensive source of supplements and/or nutraceuticals in food, a cosmetic simple and fast extraction method based on the use of cheap and relatively non-toxic solvents was developed. Pressurised liquid extraction using a green solvent was selected as an eco-friendly technique for polyphenol extraction. To select the better extraction conditions to improve recovery of phenolics compounds from artichoke by-product, the main PLE parameters (temperature and solvent) affecting extraction efficiency were optimized [37]. EtOH 50% was preliminary used as the extraction solvent and the temperature was optimised in the range of 60 °C to 120 °C. The results show a quantitative trend for polyphenols to increase proportionally with increasing temperatures from 60 °C to 100 °C; beyond this value it began to decrease (Figure 2). For these reasons, 100 °C was selected as the optimal temperature for further experiments.

The PLE technique using high pressure and temperature improves the extraction capacity of water. Consequently, the same solvent conditions used in the preliminary experiments of the USAE were tested to evaluate a possible reduction of ethanol consumption. The yields of PLE, obtained using H_2_O, EtOH and EtOH50% as the extraction solvent, were 22.6%, 12.4%, and 21.4%, respectively. To evaluate the antioxidant activity and total phenol content on different extraction conditions, spectrophotometric assays were carried out; whereas, to determine the extraction efficiency of target polyphenol compounds, quantitative analyses were carried out by UPLC–UV using the external standard method. Calibration curves in the concentration range of 1–15 μg mL^−1^ were used to quantify their content into PLE extract. The external standard calibration curves for all used standards provided good linearity within the investigated concertation range, with correlation coefficients (R^2^) ranging from 0.9940 to 0.9983, respectively, and all information were provided in (Appendix A). The result of quantitative analysis of PLE extracts are shown in Table 3. These huge datasets obtained by mixing different analytical methods were interpretated by chemometric multivariate analysis to identify the metabolomic variation among green extracts and to reveal the main metabolites responsible for the antioxidant activity.

### 2.3. Multivariate Data Analysis

Multivariate data analysis is an imaging statistical technique that is useful to easily plot large datasets for several applications in scientific fields, like control and optimisation processes [38]. In the present study, the chemical metabolome of *C. cardunculus* leaves was defined by employing different eco-sustainable solvents in a pressurised liquid extraction system.

The UPLC–UV profiles, used for the quantitative analysis of the main identified compounds, were used for a supervised multivariate data analysis approach. To this extent, multivariate data analysis was performed by measuring the selected peak area (AUC) for each metabolite identified by standard and a dataset was obtained. Data were replicated by biological and technical triplicates for each sample to confirm the statistical relevance of the model; mean value were used in the imaging plots. The resulting data matrix was used for exploratory principal component analysis (PCA) and partial least square (PLS) regression to acquire a general insight and visualise any relation (trends, outliers) among the observations (samples). The resulted models, obtained after scaling data by Pareto mode, showed plots within the covariance ellipse confirming the absence of outliers.

#### 2.3.1. Supervised Principal Component Analysis (PCA) of *C. cardunculus* Extracts

An overview of chemical variability among the solvents used for the extraction was performed by PCA. The PCA score plot allows the separation of the analysed samples into clusters, while the loading plot indicate the metabolites most influencing the separation. The choice of principal components was established based on the fitting (R2X) and predictive (Q2X) values of the model. The first component explained 81.2% of the variance while the second one explained 18.8%. Therefore, the supervised PCA score scatter plot discriminated against the different solvents used for the extraction of *C. cardunculus* by-products. The H_2_O and EtOH50% extracts clustered on the left side of the Principal Component [PC1] while the EtOH extract lay on the opposite side of the plot (Figure 3A).

The PCA loading plot showed the signals responsible for the distribution on the PCA score plot. In particular, the loadings plot shows that the clustering extracts were characterized by the main presence of chlorogenic acid, cynarin, 1-*O*-caffeoylquinic acid on the top left quadrant of the plot while the 4-*O*-caffeoylquinic acid lay alone on the bottom left quadrant (Figure 3B) representing the main specialised metabolite influencing the chemical variability of the samples. The coloured bar in Figure 3A represents the incidence of the 4-*O*-caffeoylquinic acid on the clustering of PCA model. Its occurrence is mostly significative in H_2_O extract (red spot) more than EtOH50% extract (light blue spot).

#### 2.3.2. Partial Least Square (PLS) Regression Analysis with TPC and DPPH Tests

PLS is an imaging method used to highlight the regression analysis of two data matrices obtained by different tools [39]. Chromatographic data (X matrix) was integrated to the total phenol content (TPC) and DPPH data (Y1 and Y2 matrix) with the aim to correlate the relative abundance of specific phenols affecting the total phenol content in the prepared extracts and relative radical scavenging activity.

PLS score plot with component one explained 58.7% of the variation and component two accounted for 41.3% of the variation, exhibiting a good separation between these groups. The PLS analysis showed a distinct separation (R2Y, 0.83) and good predictability (Q2, 0.54) and was validated by Hotelling’s T^2^ (95%) test to prove that the model was credible and robust.

The PLS score plot (Figure 4A) showed a good separation among the extracts. In particular, the more active EtOH50% and H_2_O extracts were separated from EtOH extracts by the first principal component [PC1].

Moreover, EtOH50% and H_2_O extracts are separated by the second principal component [PC2]. The loadings scatter plot shows the metabolites that potentially contribute to the separation observed in the score scatter model based on the TPC and DPPH activity (Figure 4B). The metabolites in the loading plot that are distant from the origin can be considered markers of the activity as a confirmation of their different distribution in the examined samples. In addition, the specific contribution of single variables to the PC1 is shown in Figure 4B. Specifically, 1-*O*-caffeoylquinic acid, cynarin, and 3,5dicaffeoylquinic acid are on the left side of the loadings plot along the PC1. Among all the specialised metabolites, the 4-*O*-caffeoylquinic acid lay on the top-left side of the PLS loadings plot and chlorogenic acid lay on the bottom-left quadrant of the model acting as good discriminants of the H_2_O and EtOH50% extracts, respectively, along the PC2. The two variables contribute to the discrimination of the H_2_O and EtOH50% extracts characterised by the higher total phenol content and better radical scavenging activity. The accurate analysis of both PCA and PLS models suggest that the EtOH 50% and H_2_O are the extracts clustering according to the occurrence of 4-*O*-caffeoylquinic acid. Moreover, the occurrence of the same specialised metabolite influences the PCA clustering of H_2_O more than EtOH50%, suggesting a possible reduction of the organic solvent on the extraction efficiency of phenolic constituents.

## 3. Materials and Methods

### 3.1. Chemicals

Analytical-grade methanol and ethanol for extractions and solvent-partition were obtained from Sigma-Aldrich (Milan, Italy). Ultrapure water (18 MΩ) was prepared by a Milli-Q purification system (Millipore, Bedford, MA, USA). LCMS-grade acetonitrile (ACN), water, and formic acid were supplied by Romil (Cambridge, UK). Reference standards (>98% HPLC grade) apigenin, luteolin, chlorogenic acid (5-caffeoylquinc acid, 5-CQA), 4-*O*-caffeoylquinic acid, cynarin (1,3-dicaffeoylquinc acid, 1,3-diCQA), 1,5-dicaffeoylquinc acid, 3,5-dicaffeoylquinc acid, 4,5-dicaffeoylquinc acid, luteolin-7-*O*-glucoside, apigenin-7-*O*-rutinoside, 1,1-diphenyl-2-picrylhydrazyl (DPPH^•^), and Folin–Ciocalteu’s phenol reagent were purchased from Sigma-Aldrich (Milan, Italy), 1-*O*-caffeoylquinic acid and 1,4-dicaffeoylquinc acid were purchased from Extrasynthase (Lyon, France). Standard stock solutions (1 mg mL^−1^) of each compound were prepared in methanol and stored at 4 °C. Diluted solutions and standard mixtures were prepared in MeOH/H_2_O 2:8, *v*/*v*.

### 3.2. Plant Sample

External leaves of *C. cardunculus* L. of wild artichoke were randomly collected, approximately 100 g, in July 2019 in Calabria region (Italy). The raw materials were freeze-dried. The dry samples were finely blended using a knife mill Grindomix GM 200 (Retsch, Haan, Germany), and the ground samples were stored at room temperature before the extraction.

### 3.3. Ultrasound-Assisted Extraction (USAE) and Pressurised Liquid Extraction (PLE)

The ground sample (1 g) was extracted in triplicate for 10 min at 25 °C with three different solvents (H_2_O, EtOH, EtOH 50%) with a matrix/solvent ratio of 1:10 *w*/*v* in an ultrasound bath (Labsonic LBS2, Treviglio, Italy). Then, the samples were centrifuged at 6000 rpm for 15 min, the supernatant was recovered, and the extracts were pooled. Finally, the extract was dried by rotavapor (strike 300, Sterogless, Perugia, Italy) and the extraction yield was calculated gravimetrically. The extracts were stored at −20 °C until the subsequent analysis.

Green extracts were prepared by pressurised liquid extraction (PLE) using an automated Dionex ASE 350 system (Dionex, Sunnyvale, CA, USA). Stainless steel cells of 5 mL were used as extraction cells while 60 mL glass vials with Teflon septa were used for extract collection. Nitrogen was supplied to assist the pneumatic system and to purge extraction cells. One gram of ground dried leaves was introduced into the extraction cell and two circular PTFE frits (Sigma-Aldrich—Saint Louis, MO, USA), 20 mm porosity, were placed at each end of stainless steel extraction cells. Different extraction conditions were performed using several solvent compositions: water, ethanol, and ethanol–water mixture (50% *v*/*v*) at different temperatures (60, 80, 100 and 120 °C); pressure, 1500 psi; static: time, 5 min; flush volume, 150%; purge, N_2_ 100 s; number of cycles, 2. After removing the organic solvent with a rotary evaporator, all extracts were adjusted to a final volume of 30 mL with distilled water. An aliquot of 1 mL was filtered (syringe-filter PTFE 0.45 mm, Phenomenex, Italy) and directly analysed by UPLC–UV/HRMS, whereas the rest of the extract was lyophilized to determine the extraction yield expressed as g PLE extract/100 g sample dry matter (DM).

### 3.4. Qualitative and Quantitative Analysis by UPLC-ESI/HRMS-UV

Qualitative and quantitative analyses were performed by using a system of liquid chromatography coupled with electrospray ionization (ESI) and high-resolution mass spectrometry (UPLC-ESI/HRMS), a Waters ACQUITY UPLC system coupled with a Waters Xevo G2-XS QTof Mass Spectrometer (Waters Corp., Milford, MA, USA), operating in negative ionisation mode. The chromatographic separations were performed on a biphenyl 100 mm × 2.1 mm, 2.6 μm column (Phenomenex, Torrance, CA, USA), by using a mobile phase consisting of 0.1% formic acid in water (*v*/*v*) as solvent A and 0.1% formic acid in acetonitrile (*v*/*v*) as solvent B, a flow rate of 0.4 mL min^−1^, and a linear gradient held at 0–2.0 min, 5–10% B; 2.0–17.0 min, 10–35% B; 17.0–18.0 min, 35–50%; 18.0–20.0 min, 50–70% B; 20.0–22.0 min, 70–95%, after each run of 5 min of wash (95% B), and 5 min of equilibration was performed before the next sample injection. The autosampler was set to inject 5 μL of each sample at concentration of 0.5 mg mL^−1^. For the ESI source, the following experimental conditions were adopted: electrospray capillary voltage 2.5 kV, source temperature 150 °C, and desolvation temperature 500 °C. MS spectra were acquired by full range acquisition covering a mass range from 50 to 1200 *m*/*z*. In order to allow HRMS/MS analysis, data-dependent scan (DDA) experiments were performed selecting the first and the second most intense ions from the HRMS scan event and submitting them to collision-induced dissociation (CID) by applying the following conditions: a minimum signal threshold at 250, an isolation width at 2.0, and normalized collision energy at 30%. Both in full and in MS/MS scan mode, resolving power of 30000 was used. Stock solutions (1 mg mL^−1^) of compounds used as external standards (ES) were prepared by dissolving each compound in a solution of methanol. Increasing ES concentrations (1, 5, 10, and 15 μg mL^−1^) were prepared for the calibration curve construction. In particular, 5 μL of each standard solution at each concentration was injected in a technical triplicate. Calibration curves were obtained by plotting the area of ES against the known concentration of each compound by using linear regression, performed using the Excel 2016 Software. Calibration curves for each standard showed good linearity (ANOVA) within the range of concentrations investigated, with correlation coefficients (R^2^) in the range of 0.994 to 0.999.

### 3.5. Determination of Total Phenol Content by Folins–Ciocalteu Assay

Total polyphenol content (TPC) of sample extract was determined by Folins–Ciocalteu assay as described by Guzzetti et al. (2021) [40] with some modifications. Briefly, the assay was performed in the cell by adding 400 μL of H_2_O milliQ, 80 μL of sample or standard, 40 μL of Folins–Ciocalteu reagent, and finally 480 μL of a 10.75% Na_2_CO_3_ solution. After a 30 min incubation time, the samples were read at a wavelength of 760 nm with a Cary 60 UV-Vis spectrophotometer (Agilent Technologies, Inc. Santa Clara, CA, USA). The samples were tested at the concentrations to 0.2 mg mL^−1^ and quantification was obtained by calibration line using gallic acid as standard at the range concentration of 0–100 μg mL^−1^. Results are expressed as mg of gallic acid equivalents for g of extract (mgGAE/gEXT).

### 3.6. Determination of DPPH Radical Scavenging Activity

The free radical scavenging activities of tested samples were measured using the 1,1-diphenyl-2-picrylhydrazyl radical (DPPH^•^) by the method previously described by Guzzetti et al. (2021) [40] with some modifications. An aliquot (50 μL) of the extract solution or standard solution (2.5–10 μg mL^−1^) was added to 950 μL of prepared DPPH^•^ solution 0,1 M. After incubation for 30 min at room temperature, samples were read at a wavelength of 515 nm. The experiments were carried out in triplicate. The DPPH^•^ results were expressed as µmol of Trolox equivalent for g of extract (µmolTrolox/gEXT).

### 3.7. Statistical and Multivariate Analysis

The results of radical scavenging activity were expressed as mean ± standard deviation (SD) of three independent experiments. All data were analysed using the software Microsoft Excel 2016 Version 16.55 (Microsoft). The analysis of variance (ANOVA) and Student’s t-test were applied to estimate statistical differences (considered to be significant at *p* ≤ 0.001). The multivariate data analysis was obtained by SIMCA^®^ 17 software (Sartorius, Goettingen, Germany), used for one-class model generation, model validation, and producing 2D plots. The PCA was run for obtaining a general overview of the variance of artichoke metabolites, and PLS was performed to obtain information on differences in the metabolite composition among species.

The dataset for PCA and PLS models was manually organized by Excel 2016, considering the identities of 11 metabolites (variables) identified by the UPLC–UV/HRMS analysis of three extraction solvents (observations) of *C. cardunculus* wild-type (WT_EtOH, WT_H_2_O and WT_EtOH 50%), and imported into SIMCA^®^ 17. Variables were mean-centered and Pareto-scaled. For reproducibility of the analysis, three extraction replicates and three replicate injections per sample extract were provided. The ‘Autofit’ function was used, and the model was described by two principal components.

## 4. Conclusions

A green chemistry approach was provided to evaluate a comparative metabolomic profile of wild *C. carduncuclus*. Ultrasound extraction technique was considered for the characterization of the metabolites profile while pressurised liquid extraction technique (PLE) was used for the green extraction approach to optimize the extraction yield and to reduce the impact on organic solvents and analysis time. The extraction conditions, temperature, and solvents used in PLE were monitored by spectrophotometric assays to determine the total phenol content and radical scavenging activity. Main phenolic compounds identified in the PLE extracts were quantified by UPLC–UV analysis highlighting the compounds 1-*O*-caffeoylquinic acid, 4-*O*-caffeoylquinic acid, cynarin, and luteolin-7-*O*-glucoside occurring in higher amounts. Subsequentially, targeted metabolomics-based MS experiments were utilized to reveal the compositional differences among artichoke extracts. In particular, the differences among the selected samples were provided by PCA plots, showing that the clustering of EtOH 50% and H_2_O samples mainly related to the variance of 4-*O*-caffeoylquinic acid content. Moreover, the EtOH 50% and H_2_O extracts were grouped in PLS plots due to the variance of 4-*O*-caffeoylquinic acid and chlorogenic acid, respectively, and showed higher total phenol content and radical scavenging activity. The general overview of the incidence of the specialised metabolite were higher in water extract, suggesting the reduction of organic solvents for the extraction of the phenolics from artichoke by-products.

In conclusion, multivariate analysis of PLE extracts shows similar behaviour of water and hydroalcoholic extracts in contrast to the results obtained by the ultrasonic technique, where the phenolic compounds and antioxidant activity were higher using organic solvents. Comparison of the two extraction techniques thus identified better extraction efficiency in the PLE technique, with increased extraction yield, reduced organic solvents and environmental impact, identifying wild thistle leaves as an interesting source of bioactive compounds for nutraceutical and cosmetic industries.

## Figures and Tables

**Figure 1 molecules-27-07157-f001:**
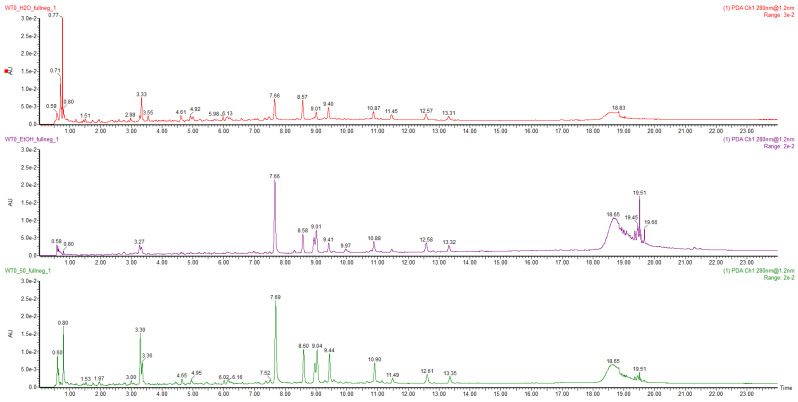
USAE chromatographic profile of H_2_O (red line), EtOH (purple line), and EtOH50% (green line) extracts (0.5 mg mL^−1^) at a wavelength of 280 nm.

**Figure 2 molecules-27-07157-f002:**
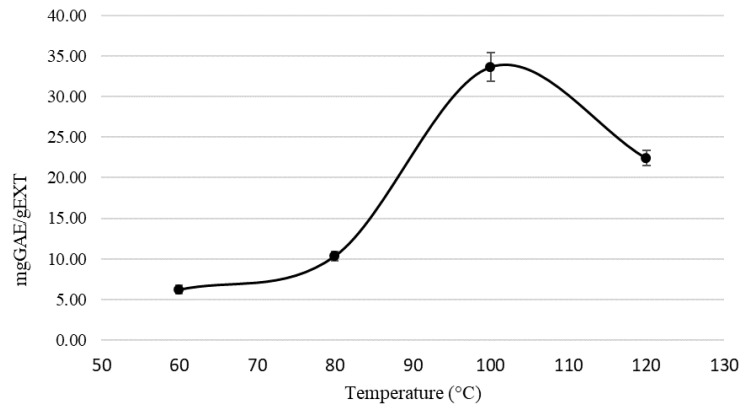
Total phenol content at different temperatures (60 °C, 80 °C, 100 °C, and 120 °C).

**Figure 3 molecules-27-07157-f003:**
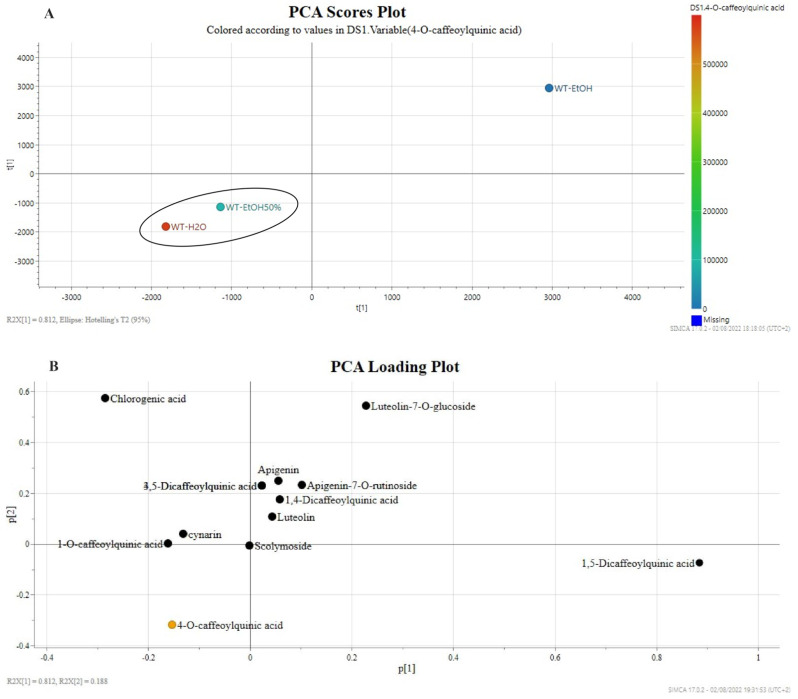
Supervised PCA regression plots of green extracts of *C. cardunculus* leaves. Scores plots (**A**) and loadings plots (**B**). WT_EtOH: ethanol extract of *C. cardunculus* wild-type; WT_H_2_O: water extract of *C. cardunculus* wild-type; WT_EtOH50%: ethanol/water (50% *v*/*v*) extract of *C. cardunculus* wild-type. The coloured bar on the right side of panel A indicates the incidence of 4-*O*-caffeoylquinic acid in the different extraction conditions.

**Figure 4 molecules-27-07157-f004:**
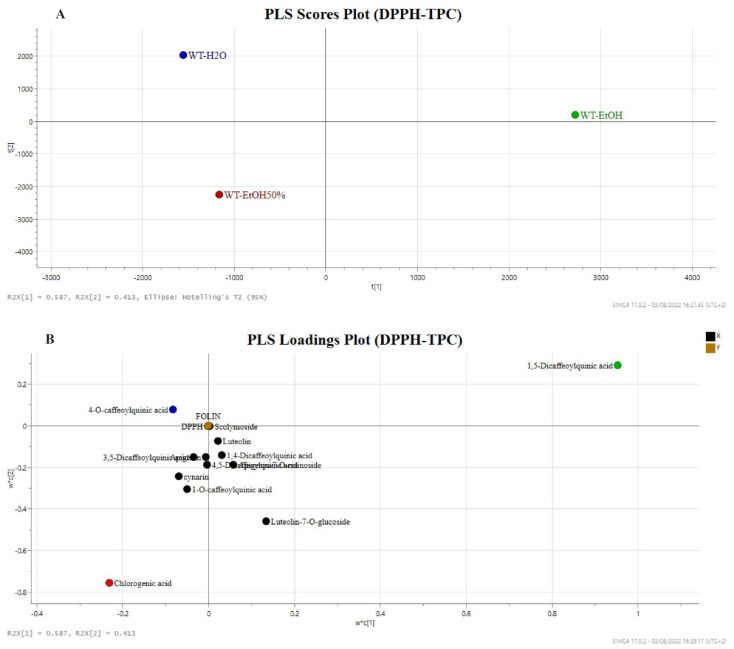
Supervised PLS regression plots of green extracts of *C. cardunculus* leaves. Scores plots (**A**) and loadings plots (**B**). WT_EtOH: ethanol extract of *C. cardunculus* wild-type; WT_H_2_O: water extract of *C. cardunculus* wild-type; WT_EtOH50%: ethanol/water (50% *v*/*v*) extract of *C. cardunculus* wild-type.

**Table 1 molecules-27-07157-t001:** Quantification of total phenols concentration and radical scavenging using Folin–Ciocalteu and DPPH assay, respectively.

Extracts	Folins–Ciocalteu Assay [mgGAE/gDM]	DPPH Assay [µmolTrolox/gEXT]
H_2_O	2.937 ± 0.054	0.159 ± 0.009
EtOH	2.499 ± 0.114	0.211 ± 0.003
EtOH50%	6.317 ± 0.161	0.246 ± 0.003

**Table 2 molecules-27-07157-t002:** Metabolites identified in the green extract of *C. cardunculus* leaves.

Peak	Tr (Minutes)	[M−H]^−^ (*m*/*z*)	Molecular Formula	Error (ppm)	MS/MS Fragments	Metabolite Identity	Reference
1	1,4	315.0716	C_13_H_15_O_9_	0.1	152.0050; 108.0170	protocatechuic acid-*O*-glycoside	[31]
2	1,6	329.0784	C_14_H_17_O_9_	−0.5	167.1225	vannilloyl-1-*O*-glycoside	Chemspider
3	2,0	353.0718	C_7_H_12_O_6_	−0.2	191.0483	1-*O*-caffeoylquinic acid	Standard
4	2,3	353.0718	C_25_H_24_O_12_	0.3	191.0483	chlorogenic acid	Standard
5	3,4	353.0718	C_32_H_36_O_18_	−1.6	191.0483	4-*O*-caffeoylquinic acid	Standard
6	4,2	471.1869	C_22_H_32_O_11_	1.2	337.0681; 191.0483	*p*-coumaric acid derivative	[32]
7	5,4	515.1197	C_25_H_24_O_12_	1.2	353.0718; 191.0483; 179.0281; 135.0378	cynarin	Standard; [33,34]
8	5,6	367.1025	C_17_H_20_O_9_	1.3	191.0483; 173.0438	3-*O*-feruloylquinic acid	[31,35]
9	6,0	681.2400	C_32_H_42_O_16_	0.0	519.1870; 357.1329; 151.0378	pinoresinol-*O*-glucopyranosyl-glycopiranoside	[36]
10	6,7	535.1806	C_26_H_31_O_12_	−0.4	357.1358	bidenilegnoside A	[31]
11	7,5	593.1511	C_27_H_30_O_15_	−0.2	447.0939; 285.0398; 151.0037	scolymoside	Standard
12	7,8	447.0932	C_21_H_20_O_11_	−0.3	357.1332; 285.0398	luteolina-7-*O*-glucoside	Standard
13	8,6	515.1197	C_25_H_24_O_12_	1.3	353.0858; 191.0561; 173.0391; 135.0378	1,4-dicaffeoylquinic acid	Standard
14	8,7	519.1870	C_26_H_32_O_12_	−0.3	357.1332; 151.0378	pinoresinol-4-*O*-glycoside	[31,36]
15	8,7	515.1197	C_25_H_24_O_12_		353.0858; 191.0561; 173.0391; 135.0378	1,5-dicaffeoylquinic acid	Standard
16	8,7	577.1570	C_27_H_30_O_12_	1.2	269.0346	apigenin-7-*O*-rutinoside	Standard; [31,34]
17	9,1	515.1197	C_25_H_24_O_12_	0.4	353.0718; 191.0483; 179.0281; 135.0378	3,5-dicaffeoylquinic acid	Standard
18	9,2	489.1036	C_23_H_22_O_12_	−0.4	447.8031; 285.0398	luteolin-7-*O*-(acetyl)-glycoside	[34]
19	9,8	515.1197	C_25_H_24_O_12_	−2.0	353.0718; 191.0483; 179.0281; 173.0391; 161.0242; 135.0378	4,5-dicaffeoylquinic acid	Standard
20	10,1	615.1359	C_29_H_27_O_15_	0.7	353.0718; 335.1522; 191.0483; 179.0281; 173.0391, 161.0242	dicaffeoyl-succinoylquinic acid	[34]
21	10,5	561.1977	C_28_H_33_O_12_	−0.1	357.1332; 342.0462; 151.0037	pinoresinol-acetilhexoside	[31]
22	11,0	269.0346	C_15_H_9_O_5_	0.2	117.0293	apigenin	Standard
23	11,6	285.0313	C_15_H_9_O_6_	0.6	151.0037; 133.0218	luteolin	Standard

**Table 3 molecules-27-07157-t003:** UPLC–UV quantitative analysis (280 and 325 nm) of different extracts (EtOH, H_2_O, and EtOH50%).

	EtOH	H_2_O	EtOH50%
1-*O*-caffeoylquinic acid	11.45 ± 0.24 ^a^	14.35 ± 0.41 ^a^	13.73 ± 0.34 ^a^
chlorogenic acid	5.13 ± 0.09 ^a^	6.51 ± 0.12 ^a^	6.21 ± 0.16 ^a^
4-*O*-caffeoylquinic acid	20.82 ± 0.34 ^a^	26.02 ± 0.35 ^a^	24.90 ± 0.61 ^a^
cynarina	79.98 ± 0.65 ^a^	103.15 ± 0.85 ^a^	98.16 ± 2.74 ^a^
scolymoside	5.11 ± 0.16 ^a^	6.24 ± 0.22 ^a^	6.00 ± 0.13 ^a^
luteolin-7-*O*-glucoside	11.99 ± 0.25 ^a^	15.21 ± 0.29 ^a^	14.51 ± 0.38 ^a^
1,4-dicaffeoylquinic acid	7.68 ± 0.11 ^a^	9.19 ± 0.19 ^a^	8.87 ± 0.18 ^a^
1,5-dicaffeoylquinic acid	3.42 ± 0.07 ^a^	4.20 ± 0.15 ^a^	4.03 ± 0.09 ^a^
apigenin-7-*O*-rutinoside	7.07 ± 0.23 ^a^	8.67 ± 0.38 ^a^	8.33 ± 0.19 ^a^
3,5-dicaffeoylquinic acid	1.31 ± 0.03 ^a^	1.62 ± 0.10 ^a^	1.55 ± 0.04 ^a^
4,5-dicaffeoylquinic acid	5.64 ± 0.31 ^a^	7.00 ± 0.06 ^a^	6.71 ± 0.16 ^a^
luteolin	3.91 ± 0.10 ^a^	4.97 ± 0.21 ^a^	4.74 ± 0.12 ^a^
apigenin	2.88 ± 0.18 ^a^	3.51 ± 0.13 ^a^	3.37 ± 0.07 ^a^
total phenol content	11.21 ± 0.70 ^b^	28.49 ± 1.53 ^b^	33.62 ± 1.73 ^b^
radical scavenging activity	36.86 ± 4.13 ^c^	105.54 ± 9.28 ^c^	144.37 ± 15.12 ^c^

^a^ Data are expressed as µg/mgEXT; ^b^ TPC expressed as gGAE/gEXT; ^c^ DPPH scavenging activity expressed as μmolTrolox/mgEXT.

## Data Availability

The original contributions presented in this study are included in the article and Appendix A, further inquiries can be directed to the corresponding author.

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
