# Peer review of "Valorisation, Green Extraction Development, and Metabolomic Analysis of Wild Artichoke By-Product Using Pressurised Liquid Extraction UPLC–HRMS and Multivariate Data Analysis"

_molecules, 2022, doi:10.3390/molecules27217157_

Round 1

Reviewer 1 Report

 Large quantities of non-edible parts of the artichoke plant (Cynara cardunculus L.) comprising leaves, stems, roots, bracts, and seeds are discarded annually during industrial processing. These by-products contain many phytochemicals such as dietary fibers, phenolic acids, and flavonoids. This paper had developed a novel valorisation strategy for the sustainable utilization of an artichoke leaves waste by combining green pressurized-liquid extraction (PLE), spectrophotometric assays and UPLC-HRMS phytochemical characterization, to obtain bioactive-rich extract with high antioxidant capacity. multivariate analysis of PLE extracts shows similar behaviour of water and gydroalcoholic extracts in contrast to the results obtained by the ultrasonic technique,where the phenolic compounds and antiocidant activity were higher using organic solvent.comparison of the two extraction techniques thus identified better extraction efficiency in the PLE technique, with increased extraction yield, reduced organic solvent and environmental impact, identifying wild thistle leaves as an interesting source of bioactive compounds for nutraceutical and cosmetic industries. 

Author Response

We would like to thank the reviewer for the positive evaluation of our manuscript, and we are pleased that the purpose of this manuscript has been achieved.

Reviewer 2 Report

This manuscript deals with extract condition, chemical composition, and antioxidant activities of artichoke by-product. The authors studied about method and solvent on extraction, and further analyzed chemical constituents in the extract. The experiments were carried out carefully, and the results were described in detail. However, polyphenols in artichoke are already reported in previous studies, and there is no finding of new components. The finding in this study is poor for publication, so I think this manuscript should be rejected.

I also point out some matters as below:

 Page 5 line 13

It has described that 50% EtOH is most appropriate solvent for extraction. However, GAE in 50% EtOH extract is lower than that of EtOH extract. What do you think? The authors should discuss about this point.

Figure 1S

Each chromatogram should have same y-axis.

 Page 7 – 8

The discussion concerning structural elucidation with MS data is unnecessary, so it should be deleted.

Author Response

We thank the reviewer for the time and effort to revise our manuscript; We are sorry for the negative evaluation of our manuscript especially regarding the novelty of our manuscript, which led the reviewer to reject it. However, the manuscript has been improved following the suggestions of reviewer and highlight the novelty compared to the paper already reported in previous studies, we hope that in the present form, the quality of manuscript has been improved enough to justify the publication.

Reviewer 3 Report

The manuscript “Valorisation, extraction optimization and metabolomic analysis of artichoke by-product using pressurized liquid extraction UPLC-HRMS and multivariate data analysis” [molecules-1968911-peer-review-v1] written by Stefania Pagliari, Ciro Cannavacciuolo, Rita Celano, Sonia Carabetta, Mariateresa Russo, Massimo Labra and Luca Campone describes an optimization to extract and isolate various natural products from Cynara cardunculus L. The authors use ultrasound assisted extraction as well as pressurized liquid extraction and optimized the procedure in particular with respect to the solvent used for extraction and the temperature. The optimization was carried out in particular with regard to the extracted amount of ingredients. UPLC-HRMS analysis was used for the quantification and quantified using calibration curves.

All experiments and calculations are performed with modern and quite common state of the art methods. The overall work seems quite well planned and performed. However, some examinations are not adequately described and thus there are a few doubts about the interpretation of the data that should be dispelled (see below). The results, however, possess some importance in furthering our knowledge of isolating natural products from food with the aim of obtaining these in quantitative terms and being able to process them further if necessary. However, there are some weaknesses in the presentation and the manuscript hence should definitely be revised (see comments).

The manuscript is of interest in the fields of Food Chemistry, Natural Product Chemistry, Phytochemistry as well as to some extent in Industrial Chemistry. However, the reviewer has some comments that should be considered by the authors before the manuscript can be accepted for publication in "Molecules".

General Comment:

a) Chapter 3.1 follows immediately after Chapter 2.7. Here a heading ("Chapter 3") and possibly some text appear to be missing in the manuscript. The authors are urgently required to check this and, if necessary, to complete it.

Scientific Comments:

b) For the quantification of the isolated substances using MS/MS, a calibration curve is created using model substances. This calibration curve should definitely be shown; e.g. in a Supplementary Material.

c) In the paragraph above Figure S2, the authors draw conclusions about the regiochemistry of the compounds they isolated based on the MS/MS measurements. It is not entirely clear to the reviewer which criteria are used to deduce the regiochemistry of the caffeoyl substituents. Authors are encouraged to elaborate on this.

d)
Ultimately, the composition of the solvents for the extraction is mainly selected using ultrasound assisted extraction and the temperature is optimized using pressurized liquid extraction. The reviewer did not notice any further significant optimization. The authors are therefore encouraged to focus mainly on these two points in the description in the text and not to speak of general "optimization".

Comments on the presentation and description

e) The authors define an "USA extraction" which the reviewer is not familiar with and about which he/she does not find anything in the current literature. The authors are therefore encouraged to explain this in some more detail, e.g. in the introduction.

f) The numbering of the tables and figures is somewhat confusing, as some are marked with "S" and others are not. The authors are encouraged to revise this and probably to transfer some tables and/or figures to a Supplementary Material.

g) The legends of the PCA figures are not very informative. The authors are encouraged to use legends to describe the figures in more detail.

Minor comments:

h) There are some inaccuracies in the typesetting that create confusion. In paragraph 2.4, three numerical values are separated by commas but not by spaces. "(1,5,10)" This is somehow confusing. The authors are therefore encouraged to critically check the entire manuscript for such (typographical) errors and to correct them so that there are no misunderstandings in the typesetting. (E.g. multiple missing dialing characters, missing commas, incorrectly abbreviated units (min instead of minutes), missing italics, etc.)

Author Response

We would like to thank the reviewer for the comments and suggestion on our manuscript. we hope that we have fully satisfied all the reviewer's requests and that in this version you are eligible for publication

Round 2

Reviewer 2 Report

I think the manuscript has been improved and modified appropriately

Reviewer 3 Report

The manuscript, now entitelt “Valorisation, green extraction development and metabolomic analysis of wild artichoke by-product using pressurized liquid extraction UPLC-HRMS and multivariate data analysis.” [molecules-1968911-peer-review-v2] written by Stefania Pagliari, Ciro Cannavacciuolo, Rita Celano, Sonia Carabetta, Mariateresa Russo, Massimo Labra and Luca Campone has been revised by the authors. The reviewer is grateful to the authors for taking to heart the previous comments and for addressing several concerns, which also were made from the other reviewers.

All comments made by reviewer 3 have been addressed in the manuscript and/or in the answer to the reviewers comments. The changes improve the equality and readability of the manuscript. The manuscript hence now fully and intelligibly an optimization to extract and isolate various natural products from Cynara cardunculus L., using ultrasound assisted extraction as well as pressurized liquid extraction. The optimization is made in particular with respect to the solvent used for extraction and the temperature to maximize
the extracted amount of ingredients. The authors now clearly present this focus in the abstract and in the revised introduction.

The manuscript should hence be acceptable for publication in Molecules.